# Synthesis and Biological Activity of Unsymmetrical Monoterpenylhetaryl Disulfides

**DOI:** 10.3390/molecules27165101

**Published:** 2022-08-10

**Authors:** Denis V. Sudarikov, Yulia V. Gyrdymova, Alexander V. Borisov, Julia M. Lukiyanova, Roman V. Rumyantcev, Oksana G. Shevchenko, Diana R. Baidamshina, Nargiza D. Zakarova, Airat R. Kayumov, Ekaterina O. Sinegubova, Alexandrina S. Volobueva, Vladimir V. Zarubaev, Svetlana A. Rubtsova

**Affiliations:** 1Institute of Chemistry, FRC “Komi Scientific Centre”, Ural Branch of the Russian Academy of Sciences, Pervomayskaya St. 48, 167000 Syktyvkar, Komi Republic, Russia; 2Department of Chemistry, Nizhny Novgorod State Technical University n.a. R.E. Alekseev, 24 Minin St., 603155 Nizhny Novgorod, Russia; 3Razuvaev Institute of Organometallic Chemistry, Russian Academy of Sciences, Tropinina 49, 603950 Nizhny Novgorod, Russia; 4Center of Collective Usage “Molecular Biology”, Institute of Biology, Komi Science Centre, Ural Branch of Russian Academy of Sciences, 28 Kommunisticheskaya Street, 167982 Syktyvkar, Russia; 5Institute of Fundamental Medicine and Biology, Kazan Federal University, 18 Kremlevskaya St., 420008 Kazan, Russia; 6Saint Petersburg Pasteur Research Institute of Epidemiology and Microbiology, 14 Mira St., 197101 Saint Petersburg, Russia

**Keywords:** unsymmetrical, monoterpenylhetaryl disulfides, antioxidant, antibacterial, antifungal activity, cytotoxicity, mutagenicity

## Abstract

New unsymmetrical monoterpenylhetaryl disulfides based on heterocyclic disulfides and monoterpene thiols were synthesized for the first time in 48–88% yields. Hydrolysis of disulfides with fragments of methyl esters of 2-mercaptonicotinic acid was carried out in 73–95% yields. The obtained compounds were evaluated for antioxidant, antibacterial, antifungal activity, cytotoxicity and mutagenicity.

## 1. Introduction

Disulfide bonds play an important role in living organisms. In the formation of the tertiary structure of proteins, covalent disulfide bonds provide the formation of cross-links that are much stronger than hydrophobic interactions or hydrogen bonds. The biochemical significance of the disulfide bond is also determined by the unique nature of the thiol-disulfide system, in which the S-S bond can be formed under conditions characteristic of biological processes—by disulfide exchange with the participation of glutathione [1]. The reactivity of disulfides is determined by the lability of the S-S bond and its ability to be cleaved by both nucleophiles and electrophiles [1].

In the presence of various thiophilic nucleophiles, disulfides behave as electrophilic reagents, which leads to heterolytic cleavage of the S-S bond. Intracellular nucleophiles such as glutathione and thioredoxins, for example, can rapidly react with di- and trisulfide compounds, disrupting cellular homeostasis [2,3].

Some biologically active lipophilic disulfides of natural origin are known. These include the symmetrical diallyl disulfide (DADS), ajoene, vinyldithiins, and diallyl polysulfides found in *Allium sativum*. DADS has moderate antioxidant activity at low concentrations (20 µM) [4], however, at higher concentrations (50–100 µM), DADS exhibits pro-oxidant activity and cytotoxicity in vitro through the accumulation of reactive oxygen species (ROS) in cells [5], which is consistent with studies on the antitumor activity of DADS on human neuroblastoma cells SH-SY5Y [6], as well as on human promyelocytic leukemia cell line HL-60 [7], leading to their selective apoptosis. The antimicrobial activity of DADS and diallyl polysulfides and the ability to inhibit the formation of biofilms on surfaces are due to the modification of thiol-containing proteins in *Pseudomonas aeruginosa*, which leads to a decrease in the level of glutathione, induction of protein aggregation and inactivation of enzymes responsible for the vital activity of microorganisms, as well as inactivation of quorum-sensing genes [4,8]. DADS prevents infection with methicillin-resistant *Staphylococcus aureus* in diabetic mice [9], has a synergistic effect with gentamicins on *Escherichia coli* [10,11] and inhibits *Helicobacter pylori* activity both in vitro and in vivo [12,13]. DADS-loaded 140 ± 30 nm niosomes effectively eliminated the fungal burden in mice infected with *Candida albicans* and increased the survival rate of infected specimens [14]. DADS has also been reported to inhibit the growth of *Aspergillus versicolor* and its toxic metabolites [15]. DADS has been found to inhibit the proliferation of HIV-1-infected cells [16]. In addition, DADS had anti-inflammatory and antioxidant effects in the dengue virus study, reducing symptoms and disease severity [17].

The symmetrical *Psammaplin A* disulfide isolated from the sponge *Psammaplysilla* and containing bromotyrosine fragments and its unsymmetrical analogs, in particular UVI5008, exhibit in vitro activity against a methicillin-resistant strain of *Staphylococcus aureus* (MRSA) [18,19]. Isolated from *Polycitorella mariae*, a new cytorellamine disulfide containing heterocyclic bromoindole fragments has significant antimicrobial activity [20]. Cytorellamine disulfide (originally a misattributed sulfide structure) has demonstrated strong antibacterial activity against *Staphylococcus aureus*, *Bacillus subtilus*, and *Escherichia coli*, as well as cytotoxicity against certain cancer cell lines [21].

Disulfiram, a drug originally used for the treatment of alcohol dependence, and unsymmetrical disulfiram-based disulfides with alkyl and aryl fragments have antibacterial activity against a large selection of gram-positive (such as *Staphylococcus aureus*—MRSA, VISA, VRSA) and gram-negative pathogenic bacteria (*Bacillus anthracisa*, *Bacillus cereus*, *Borrelia burgdorferi*), and have a synergistic effect together with vancomycin [22,23,24]. Antimicrobial therapy in a limited number of patients has shown the effectiveness of disulfiram in the treatment of borreliosis caused by *Borrelia burgdorferi* in clinical settings [25]. The authors attribute the mechanism of action of disulfiram and its derivatives to reactions with thiophilic residues in proteins, metabolites, and cofactors, as well as to a decrease in the content of cellular glutathione [22].

Synthetic alkylaryl and dialkyl disulfides show potent antimicrobial activity against *Staphylococcus aureus* and *Bacillus anthracis*, including MRSA, through inhibition of the β-ketoacyl acyl carrier protein synthase III, or FabH, a key enzyme in type II fatty acid biosynthesis [26]. The highest antimicrobial activity in vitro was found in unsymmetrical disulfides with nitrophenyl substituents, which may be the result of electronic activation of the leaving arylthio group for a nucleophilic attack on the disulfide bond [26].

Unsymmetrical disulfides with a 1,2,4-triazine substituent exhibit in vitro low micromolar inhibitory properties of MCF-7 human breast cancer cells. Disulfides with methyl and methoxy substituents in the aromatic ring demonstrated activity superior to chlorambucil [27].

Unsymmetrical disulfides are a promising platform for the development of new drugs against SARS. Unsymmetrical aromatic disulfides, incl. with fragments of substituted imidazole, pyrimidine, oxadiazole, etc., proved to be in vitro inhibitors of the protease of the SARS CoV M^pro^ coronavirus, the main causative agent of atypical pneumonia [28,29].

Semi-synthetic symmetrical and asymmetric disulfides with fragments of mono- and sesquiterpenoids demonstrate antioxidant, membrane-protective, and antiviral properties [30,31,32].

It is interesting to develop the chemistry of disulfide compounds based on natural compounds in order to study their biological properties. Combining in one molecule terpene and heterocyclic fragments connected by an S-S bond can promote the growth of already existing biological activity.

## 2. Results and Discussion

### 2.1. Synthesis of Unsymmetrical Monoterpenylhetaryl Disulfides

In our work, we synthesized a series of unsymmetrical disulfides combining monoterpene and heterocyclic fragments, and evaluated the antibacterial, antifungal, antiviral, and antioxidant activities of the synthesized conjugates.

As a heterocyclic component, we used symmetrical dihetaryl disulfides with benzothiazole, pyrimidine, and pyridine fragments, in which the disulfide bond is easily cleft in the reaction with SO_2_Cl_2_ to form sulfenyl chlorides [33,34]. The above compounds are highly reactive and often unstable, so sulfenyl chlorides were immediately involved in reactions with terpenthiols without prior isolation to form the corresponding monoterpenylhetaryl disulfides **(1–4)a–c** in 48–88% yields (Figure 1), according to the procedure [35,36].

The structure and composition of disulfides **(1–4)a–c** have been proved by IR, NMR spectroscopy and mass spectrometry. The ^1^H and ^13^C NMR spectra contain signals of both terpene and heterocyclic fragments. The structure and configuration of **1c** was confirmed by single-crystal X-ray diffraction (Figure 1). The SSHet-fragmet in the molecule **1c** is almost flat. The average deviation of non-hydrogen atoms from the plane is 0.044 Å. The cyclohexane ring adopts a chair conformation. The methyl and isopropyl groups are in equatorial positions, while the SSHet-fragment occupy axial positions. The S-S distance in **1c** is 2.0238(7) Å and is in good agreement with previously published related disulfides [37].

To assess the effect of the ester and carboxyl groups on the biological activity of disulfides for derivatives of 2-mercaptonicotinic acid, hydrolysis of disulfides **1c–4c** having an ester group was carried out. The esters were hydrolyzed in the LiOH–H_2_O–THF system. As a result, the corresponding acids **1d–4d** were obtained in 73–95% yields (Figure 2).

The structure and composition of disulfides **1d–4d** with a fragment of 2-mercaptonicotinic acid were proved by IR, NMR spectroscopy and mass spectrometry. In the ^1^H and ^13^C NMR spectra of acids **1d–4d**, there are no signals of methyl groups compared to the starting esters **1c–4c**. The mass spectra contain peaks of the corresponding molecular ions.

The synthesized disulfide conjugates were evaluated for antioxidant, antibacterial, antifungal and antiviral activities.

### 2.2. Antioxidant Activity in a Non-Cellular Model System (Brain Homogenates)

In a non-cellular model system containing easily oxidized brain lipids (Figure 2), compounds **3a–c**, **4a–c** and **3d** containing a hydroxyl group in the pinane fragment, in general, significantly more actively inhibit the oxidation of unsaturated fatty acids of brain lipids than similar in structure compounds **1a–d**, **2a–d**. If we focus on the structure of the heterocyclic substituent, then in all four groups the most active compounds contain a fragment of the methyl ester of 2-mercaptonicotinic acid (**1–4c**).

The highest antioxidant activity in this model system is demonstrated by the disulfides **3c** and **4c**, which combine a hydroxyl group in the pinane and an ester group in the heterocyclic fragment (Figure 2).

### 2.3. Erythrotoxicity, Membrane-Protective and Antioxidant Activity (on Erythrocytes)

A study of hemolytic activity showed that at a concentration of 100 μM, 13 out of 16 disulfides exhibit high erythrotoxicity (Figure 3A). Low hemolytic activity at the indicated concentration was noted only for disulfides **1a**, **2a**, and **3d**. No significant hemolytic activity was detected at a concentration of 1 μM (Figure 3B).

The membrane-protective properties of the compounds were evaluated in the model of oxidative hemolysis of erythrocytes at a concentration of 1 μM in order to avoid the effect of hemolytic activity. It was shown (Figure 4) that even at such a low concentration, individual disulfides have a statistically significant membrane-protective activity comparable to the activity of Trolox, which follows from a decrease in the death of erythrocytes in their presence under the influence of H_2_O_2_. The most active are pinane derivatives **2a**, **2b**, **2c**, **3b**, **3c**, **4a**, **4b**, **4c** and **4d**, most of which contain a hydroxyl group in the terpene moiety (**3b**, **3c**, **4a**, **4b**, **4c** and **4d**).

Thus, in both test systems used, the antioxidant activity of the obtained derivatives depended mainly on the structure of the terpene fragment, the presence of a hydroxyl group and a heterocyclic fragment in it.

### 2.4. Antimicrobial Activity

The antimicrobial activity of all compounds was evaluated on *S. aureus ATCC 29213* (MSSA), *S. aureus MRSA* clinical isolate, *P. aeruginosa ATCC 27853* and fluconazole-sensitive *Candida albicans* 703 clinical isolate. As could be seen from Table 1, compounds **1a–c** and **2b** exhibited antibacterial activity on both *S. aureus* and *P. aeruginosa* while the antifungal activity was low. By contrast, compounds **3d** and **4d** were active against yeast on low antibacterial activity background.

Unsymmetrical disulfides **1a–d** with a neomentane fragment showed antimicrobial activity against both *S. aureus* strains with MICs of 16–32 μg/mL. The antimicrobial properties of disulfides with pipane substituents varied depending on the structure of the heterocyclic fragment. Among the disulfides **2a–d**, compound **2b** with a pyrimidine substituent (MIC 16 μg/mL) and **2d** (MIC 16 μg/mL) showed the highest activity against the MSSA isolate. Compound **2a** with a benzothiazole fragment had weak activity (MIC 128 μg/mL), and **2c** was inactive against MSSA, while showing activity in the case of MRSA. The introduction of an OH group into the structure of the terpene fragment leads to an almost complete loss of the antibacterial properties of thiotherpenoids **4a–4c**. Unsymmetrical pinane disulfides **3a–c**, having a hydroxymethyl group, were similar in their biological properties to neomentane thiotherpenoids **1a–c** and showed antibacterial activity with an MIC of 32–34 µg/mL. These thiotherpenoids tested showed similar properties against the clinical methylicin-resistant *S. aureus* isolate.

All disulfides **1a–c** with the neomentane structure of the terpene part of the molecule, as well as compounds **2a** and **3a** with a benzothiazole substituent, and pinane disulfide **2b** with a pyrimidine fragment repressed the growth of *P. aeruginosa* ATCC 27853. All other compounds were inactive against this microorganism.

None of the tested asymmetric disulfides **(1–4)a–c** had antifungal activity against *Candida albicans* 703 clinical isolate. However, the presence of a carboxyl group in the 2-mercaptonicotinic acid fragment of disulfides **1–4d** led to the appearance of antifungal activity (MIC 16–128 μg/mL). 

In general, the synthesized asymmetric monoterpenylhetaryl disulfides **(1–4)a–d** have a high cytotoxicity (CC_50_) on the nuclear cells of the embryonic bovine lung (EBL). The least toxicity was shown by pinane disulfides **2a**, **2c** and **3d**. Notably, neomentane disulfides **1a** and **1b** showed mutagenicity in the Ames test on *Salmonella typhimurium* TA100 and TA102 strains, respectively, causing point mutations and reversions [38].

Thus, unsymmetrical disulfides containing monoterpene and heterocyclic substituents have antimicrobial properties against clinically significant pathogens. However, these compounds demonstrate relatively high to moderate cytotoxicity, thus reducing the possibility of further use of these compounds as potential antibiotics for systemic application, while can be used as topical agents and can be considered as a start point for the search among unsymmetrical monoterpenylhetaryl disulfides for new substances with low toxicity and selectivity against pathogenic strains of microorganisms.

### 2.5. Antiviral Activity

The synthesized hybrid compounds were screened for antiviral activity. The cytotoxicity of the compounds (CC_50_) was evaluated in uninfected MDCK cells (canine kidney cells). The antiviral activity (IC_50_) of the synthesized substances was studied on the model of influenza infection of MDCK cells caused by the influenza virus strain A/Puerto Rico/8/34 (H1N1), which is resistant to adamantane-type antiviral drugs-amantadine and remantadine. The obtained data were used to calculate the value of the selectivity index (SI) for each of the tested samples of monoterpenylhetaryldisulfides. The results of the experiments performed are shown in the Table 2.

All compounds, with the exception of disulfides **3c**, **3d** and **4d**, showed high virus-inhibiting properties. However, almost all of them showed significant cytotoxicity.

Based on the totality of data and taking into account the highest value of the selectivity index (SI > 10) [39], disulfide **2d** with the pinane structure of the terpene and heterocyclic fragment of nicotinoate can be attributed to the active compounds.

Importantly, the virus used in our study is resistant to the reference compound Rimantadine having, similarly to the compound library under investigation, cage fragment in its structure. This is due to amino acid substitution S31N in transmembrane domain of viral proton pump M2 which is target for adamantane-based anti-influenza drugs, Rimantadine and Amantadine. The activity of **2d** appeared higher than that of Rimantadine. The results obtained in our study, therefore, suggest that either **2d** is able to block M2 channel of rimantadine-resistant virus or, otherwise, it has another target and an alternative mechanism of activity. Further studies are therefore needed to decipher the exact mode of anti-viral action of this class of compounds.

## 3. Conclusions

Thus, new unsymmetrical monoterpenylhetaryl disulfides based on heterocyclic disulfides and monoterpene thiols have been synthesized for the first time. The described disulfides, in general, have good indicators of antioxidant, antimicrobial and antiviral activity. However, given the high in vitro cytotoxicity in different models, the possibility of further systemic use of these compounds as potential antioxidants, antibiotics, and antiviral agents is limited, their chemotype can be considered as a start point for the among unsymmetrical monoterpenylhetaryl disulfides for new substances with low toxicity and selectivity for pathogenic strains of microorganisms and viruses. In addition, cytotoxicity analysis shows differences in the activity of these compounds in different models (mice erythrocytes, embryonic bovine lung cells, and canine kidney cells), which may indicate a selective effect on different cell types and provides prerequisites for testing them for antitumor activity.

## 4. Materials and Methods

### 4.1. General Information

IR spectra were registered on a Shimadzu IR Prestige 21 infrared Fourier spectrometer (Shimadzu, Duisburg, Germany) in a thin layer or in KBr tablets. ^1^H NMR and ^13^C NMR spectra were recorded on a Bruker Avance 300 spectrometer (Bruker BioSpin, Ettlingen, Germany) (300.17 and 75.48 MHz) in C_6_D_6_, CDCl_3_ and acetone-*d6* using the signal of the indicated solvent as an internal standard (See Appendix A). ^13^C NMR spectra were registered in the *J*-modulation mode. The complete assignment of ^1^H and ^13^C signals was performed using 2D homo- (^1^H–^1^H COSY, ^1^H–^1^H NOESY) and heteronuclear experiments (^1^H–^13^C HSQC, ^1^H–^13^C HMBC). For convenience of interpretation of the NMR spectra on the structures **(1–4)a–d**, the numbering of carbon atoms is marked, which may not correspond to the recommended IUPAC (See Figure 1 and details of the NMR spectra in the experimental section). Mass spectra were recorded on a Thermo Finnigan LCQ Fleet instrument (Thermo Fisher Scientific, Waltham, USA) equipped with an MS detector. Masses were scanned in the *m*/*z* range 50–2000 (ESI, 20 eV). Melting points were determined on a Sanyo Gallenkamp MPD350BM3.5 instrument (Sanyo Gallenkamp, Southborough, UK) and were not corrected. The angles of optical rotation were measured on an automated digital polarimeter Optical Activity PolAAr 3001 (Optical Activity, Huntingdon, UK). Sorbfil plates were used for thin-layer chromatography; the visualizing agent was a solution of phosphoromolybdic acid in ethanol. Alfa Aesar silica gel (0.06–0.2 mm) was used for column chromatography. The commercially available 2,2′-dithiobisbenzothiazol, 99% (Sigma Aldrich, St. Louis, MO, USA), 4,6-dimethylpyrimidine-2-thiol, 98% (Alfa Chemistry, Heysham, UK), 2-mercaptonicotinic acid, 90% (Alfa Aesar, Heysham, UK) were used without additional purification.

The diffraction data for compound **1c** were collected on a Bruker D8 Quest diffractometer (Mo-K_α_ radiation, ω-scan technique, λ = 0.71073 Å). The intensity data were integrated by the SAINT [40] program. The structure was solved by dual methods [41] and was refined on $F_{\mathrm{hkl}}^{2}$ using the SHELXTL package [42]. All non-hydrogen atoms were refined anisotropically. All hydrogen atoms were placed in calculated positions and were refined using a riding model (U_iso_(H) = 1.5U_eq_(C) for CH_3_ groups and U_iso_(H) = 1.2U_eq_(C) for other groups). The SADABS program [43] was used to perform absorption corrections. The disorder of C(15) and C(16) methyl groups was modeled with EADP/SADI/ISOR constraints. CCDC 2182740 contains the supplementary crystallographic data. These data can be obtained free of charge from The Cambridge Crystallographic Data Centre via https://www.ccdc.cam.ac.uk/structures, accessed on 28 July 2022.

### 4.2. General Procedure

*2,2′-Disulfanediyldinicotinic acid.* To a solution of 8 g (0.2 mol) of sodium hydroxide in 200 mL of water 31 g (0.2 mol) of 2-mercaptonicotinic acid was added with stirring. A 7% aqueous solution of iodine with potassium iodide (1:1 eq) was added to the resulting solution with vigorous stirring until the color of iodine ceased to disappear. The precipitate formed was filtered off and thoroughly washed with water and then dried. The dried precipitate was boiled 2 times for 30 min in 100 mL of methylene chloride and filtered off. The yield of the disulfide is 27.4 g (89%) with m.p. 222–224 °C. The spectral data correspond to the literature data [44].

*1,2-bis(4,6-dimethylpyrimidin-2-yl)disulfide* was prepared similarly, m.p. 167–168 °C, spectral characteristics correspond to the literature [45].

*Dimethyl 2,2′-disulfanediyldinicotinate.* To a suspension of 6.16 g (20 mmol) of 2,2′-disulfanediyldinicotinic acid in 20 mL of absolute methanol was added 5 g of concentrated sulfuric acid and boiled for 5 h. Then, the methanol was distilled off in a vacuum, the residue was treated with water and filtered off. The filtrate was cooled to 0 °C and carefully neutralized to a slightly alkaline reaction (pH 8–9) with a NaHCO_3_ solution. The precipitate formed was filtered off and dried. The dried precipitate was dissolved in 10 mL of methylene chloride, the solution was filtered. The filtrate was evaporated in a vacuum, and the residue was boiled 2 times for 30 min in 30 mL of diethyl ether and filtered off. The yield of disulfide is 3.7 g (55%), m.p. 199–200 °C. ^1^H-NMR (300 MHz, CDCl_3_): 4.01 (s, 3H, OCH_3_), 7.10 (m, 1H), 8.23 (d.d, 1H, ^3^*J* = 7.6, *J* = 1.3), 8.48 (d.d., 1H, ^3^*J* = 5.6, *J* = 1.3). IR (KBr), cm^−1^: 496; 825, 765, 760, 752, 731 (ρ^Ar^_CH_); 1287, 1246, 1136, 1067 (*ν*_CN,CS_), (β^Ar^_CH_); 1576, 1557, 1396 (*ν*^Ar^_CH_); 1713 (*ν*
_C=O_).

Neomenthanthiol (**1**) [46], (−)-cis-myrthanethiol (**2**) [47], 10-hydroxyisopinocampheylthiol (**3**) [48] and 10-sulfanylisopinocampheol (**4**) [49] were synthesized according to known methods.

#### 4.2.1. General Procedure for the Synthesis of Unsymmetrical Disulfides

The procedure is based on methods for the synthesis of sulfenyl chlorides [33,34] and their interaction with thiols [35,36].

A total of 0.45 mmol (1 equiv.) of dihetaryl disulfide was dissolved in 10 mL of CH_2_Cl_2_ and cooled on ice-bath. A solution of 0.68 mmol (1.5 equiv.) SO_2_Cl_2_ in CH_2_Cl_2_ (*v*/*v* 1:10) was added to the cooled mixture and stirred at 0 °C for 5 min. The solvent and the excess of unreacted SO_2_Cl_2_ were removed under reduced pressure. The residue was dissolved in 5 mL of CH_2_Cl_2_, and 0.90 mmol (2 equiv.) of terpenthiol in 4 mL of CH_2_Cl_2_ was added at 0 °C. After 5–10 min, the ice-bath was removed and left to stir at room temperature. The progress of the reaction was monitored by TLC using the eluents indicated in the descriptions of the compounds. Upon completion of the reaction, the mixture was diluted with 15–20 mL of CH_2_Cl_2_ and washed with brine, the organic layer was dried over Na_2_SO_4_, and the solvent was removed under vacuum. The products were isolated by silica gel column chromatography using the same eluents as for TLC.

*2-(((1S,2S,5R)-2-isopropyl-5-methylcyclohexyl)disulfanyl)benzo[d]thiazole (***1a***).* Yield: 61%. Yellowish oil. *R*_f_ = 0.5 (PhH/EtOAc, 20:1). ${\left[\alpha \right]}_{D}^{26}$ = +96.9 (*c* = 1.0, CHCl_3_). IR (KBr): 2951, 2918, 1458, 1427, 1236, 1003, 756, 725, 671. ^1^H-NMR (300 MHz, C_6_D_6_): 0.72–0.83 (m, 1H, H-4a), 0.79 (d, 3H, *J* = 6.6, Me-7), 0.87 (d, 3H, *J* = 6.6, Me-10), 0.98–1.12 (m, 2H, H-2, H-6a), 0.99 (d, 3H, *J* = 6.6, Me-9), 1.10–1.24 (m, 1H, H-3a), 1.60–1.73 (m, 2H, H-3b, H-4b), 1.70 (ddt, 1H, *J* = 13.0, 10.0, 6.6, H-8), 1.81–2.00 (m, 1H, H-5), 2.30 (dq, 1H, *J =* 13.9, 2.9, H-6b), 3.52–3.57 (m, 1H, H-1), 6.92 (t, 1H, *J* = 7.7, H-5′), 7.07 (t, 1H, *J* = 7.8, H-4′), 7.31 (d, 1H, *J* = 7.8, H-6′), 7.88 (d, 1H, *J* = 7.8, H-3′). ^13^C-NMR (75 MHz, CDCl_3_): 21.6 (C-10, C-9), 22.5 (C-7), 26.4 (C-3), 27.0 (C-5), 30.7 (C-8), 35.7 (C-4), 40.1 (C-6), 49.4 (C-2), 55.5 (C-1), 121.6 (C-6′), 123.1 (C-3′), 125.0 (C-5′), 126.0 (C-4′), 136.9 (C-7′), 156.2 (C-2′), 173.9 (C-1′). MS (ESI, 20 eV): 338.20 (100) [M+H]^+^. M = 337.56, C_17_H_23_NS_3_.

*2-(((1S,2S,5R)-2-isopropyl-5-methylcyclohexyl)disulfanyl)-4,6-dimethylpyrimidine (***1b***).* Yield: 62%. Light-brown crystal. *R*_f_ = 0.39 (PhH/EtOAc, 20:1). ${\left[\alpha \right]}_{D}^{23}$ = +135.7 (*c* = 0.8, CHCl_3_). IR (KBr): 2949, 2916, 1581, 1531, 1446, 1369, 1256, 1028, 880, 849, 546. ^1^H-NMR (300 MHz, C_6_D_6_): 0.86–0.93 (m, 1H, H-4a), 0.99 (d, 3H, *J* = 6.6, Me-7), 1.03 (d, 3H, *J* = 6.6, Me-10), 1.09–1.25 (m, 2H, H-2, H-6a, H-2), 1.38 (d, 3H, *J* = 6.6, Me-9), 1.42–1.56 (m, 1H, H-3a), 1.75–1.88 (m, 2H, H-3b, H-4b), 2.06 (ddt, 1H, *J* = 13.2, 10.3, 6.6, H-8), 2.18 (s, 6H, Me-5′, Me-6′), 2.32–2.49 (m, 1H, H-5), 2.63 (dq, 1H, *J =* 13.6, 2.8, H-6b), 3.80–3.84 (m, 1H, H-1), 6.08 (s, 1H, H-3′). ^13^C-NMR (75 MHz, CDCl_3_): 21.7 (C-10), 21.9 (C-9), 22.8 (C-7), 23.8 (C-5′, 6′), 26.7 (C-3), 26.9 (C-5), 30.9 (C-8), 36.1 (C-4), 39.9 (C-6), 49.6 (C-2), 53.6 (C-1), 117.0 (C-3′), 167.7 (C-2′, 4′), 172.2 (C-1′). MS (ESI, 20 eV): 311.15 (100) [M+H]^+^. M = 310.52, C_16_H_26_N_2_S_2_.

*Methyl 2-(((1S,2S,5R)-2-isopropyl-5-methylcyclohexyl)disulfanyl)pyridine-3-carboxylate (***1c***).* Yield: 80%. Colorless crystals. M.p. 96 °C. *R*_f_ = 0.26 (PhH). ${\left[\alpha \right]}_{D}^{26}$ = +119.8 (*c* = 1.0, CHCl_3_). IR (KBr): 2951, 2912, 1715 (C=O), 1557, 1445, 1398, 1288, 1240, 1130, 1068, 760. ^1^H-NMR (300 MHz, C_6_D_6_): 0.82–0.96 (m, 1H, H-4a), 0.86 (d, 3H, *J* = 6.6, Me-7), 0.93 (d, 3H, *J* = 6.6, Me-10), 0.99–1.15 (m 2H, H-2, H-6a), 1.30 (d, 3H, *J* = 6.6, Me-9), 1.37–1.51 (m 1H, H-3a), 1.64–1.77 (m 2H, H-3b, H-4b), 2.03 (dq, 1H, *J* = 9.5, 6.6, H-8), 2.23–2.39 (m, 1H, H-5), 2.48 (dq, 1H, *J =* 13.8, 2.7, H-6b), 3.37 (s, 3H, Me-7′), 3.73–3.78 (m, 1H, H-1), 6.38 (dd, 1H, *J* = 7.7, 4.8, H-3′), 7.75 (dd, 1H, *J* = 7.7, 1.7, H-4′), 8.31 (dd, 1H, *J* = 4.8, 1.7, H-2′). ^13^C-NMR (75 MHz, CDCl_3_): 21.6 (C-9), 21.9 (C-10), 22.8 (C-7), 26.7 (C-3), 27.0 (C-5), 30.9 (C-8), 36.2 (C-4), 40.1 (C-6), 49.9 (C-2), 52.2 (C-6′), 52.9 (C-1), 120.0 (C-3′), 124.6 (C-5′), 138.9 (C-4′), 152.4 (C-2′), 163.6 (C-1′), 165.7 (C-6′). MS (ESI, 20 eV): 340.09 [M+H]^+^. M = 339.51, C_17_H_25_NO_2_S_2_. A colorless prismatic crystal of the orthorhombic system had size 0.47 × 0.44 × 0.25 mm, space group *P*2_1_2_1_2_1_, *a* = 8.9974(4), *b* = 13.2611(5), *c* = 15.5898(6) Å, *α* = *β* = *γ* = 90°, *V* = 1860.10(13) Å^3^, Z = 4, μ = 0.292 mm^−1^, *d*_calc_ = 1.212 g/cm^3^, F(000) = 728. A dataset of 27448 reflections was collected at scattering angles 2.016° < θ < 26.007°, of which 3659 were independent (R_int_ = 0.0215), including 3421 reflections with *I* > 2σ(*I*). The final refinement parameters were R_1_ = 0.0301, wR_2_ = 0.0687 (all data), R_1_ = 0.0267, wR_2_ = 0.0664 [*I* > 2σ(*I*)] with GooF = 1.069. Δρ*_e_* = 0.122/−0.215 *e* Å^−3^; Flack parameter = −0.039(11).

*2-((((1S,2R,5S)-6,6-dimethylbicyclo [3.1.1]heptan-2-yl)methyl)disulfaneyl)benzo[d]thiazole (***2a***).* Yield: 71%. Yellowish oil. *R*_f_ = 0.26 (PE/EtOAc 50:1). ${\left[\alpha \right]}_{D}^{24}$ = −75.7 (*c* = 0.3, CHCl_3_). IR (KBr): 2983, 2911, 1462, 1427, 1308, 1234, 1078, 1003, 756, 725, 667. ^1^H-NMR (300 MHz, CDCl_3_): 0.94 (d, 1H, *J =* 10.2, H-7a), 1.00 (s, 3H, Me-8), 1.20 (s, 3H, Me-9), 1.50–1.65 (m, 1H, H-3a), 1.86–2.15 (m, 5H, H-5, H-4a, H-4b, H-1, H-3b), 2.33–2.52 (m, 2H, H-2, H-7b), 2.99–3.12 (m, 2H, H-10a, H-10b), 7.33 (t, 1H, *J* = 7.8, H-5′), 7.44 (t, 1H, *J* = 7.8, H-4′), 7.82 (d, 1H, *J* = 7.8, H-6′), 7.87 (d, 1H, *J* = 7.8, H-3′). ^13^C-NMR (75 MHz, CDCl_3_): 21.7 (C-3), 23.3 (C-8), 26.0 (C-4), 27.8 (C-9), 33.1 (C-7), 38.7 (C-6), 40.3 (C-2), 41.1 (C-5), 45.3 (C-1), 46.8 (C-10), 121.1 (C-6′), 122.1 (C-3′), 124.4 (C-5′), 126.2 (C-4′), 135.8 (C-7′), 155.2 (C-2′), 173.2 (C-1′). MS (ESI, 20 eV): 336.24 (100) [M+H]^+^. M = 335.54, C_17_H_21_NS_3_.

*2-((((1S,2R,5S)-6,6-dimethylbicyclo[3.1.1]heptan-2-yl)methyl)disulfanyl)-4,6-dimethylpyrimidine (***2b***).* Yield: 57%. Light-brown powder. M.p. 45 °C. *R*_f_ = 0.34 (PE/EtOAc 10:1). ${\left[\alpha \right]}_{D}^{24}$ = −76.4 (*c* = 0.3, CHCl_3_). IR (KBr): 2938, 2913, 1582, 1531, 1435, 1340, 1256, 1032, 880, 764. ^1^H-NMR (300 MHz, CDCl_3_): 0.88 (d, 1H, *J =* 9.5, H-7a), 0.99 (s, 3H, Me-8), 1.19 (s, 3H, Me-9), 1.52–1.66 (m, 1H, H-3a), 1.80–2.14 (m, 5H, H-5, H-4a, H-4b, H-1, H-3b), 2.29–2.47 (m, 2H, H-2, H-7b), 2.46 (s, 6H, Me-5′, Me-6′), 2.87–3.03 (m, 2H, H-10a, H-10b), 6.78 (s, 1H, H-3′). ^13^C-NMR (75 MHz, CDCl_3_): 21.8 (C-3), 23.2 (C-8), 23.9 (C-5′, C-6′), 26.1 (C-4), 27.9 (C-9), 33.3 (C-7), 38.6 (C-6), 39.9 (C-2), 41.2 (C-5), 45.4 (C-1), 46.0 (C-10), 116.9 (C-3′), 167.6 (C-2′,C-4′), 170.8 (C-1′). MS (ESI, 20 eV): 309.30 (100) [M+H]^+^. M = 308.50, C_16_H_24_N_2_S_2_.

*Methyl 2-((((1S,2R,5S)-6,6-dimethylbicyclo[3.1.1]heptan-2-yl)methyl)disulfaneyl)pyridine-3-carboxylate (***2c***).* Yield: 74%. Yellowish oil. *R*_f_ = 0.24 (PE/EtOAc 10:1). ${\left[\alpha \right]}_{D}^{24}$ = −63.7 (*c* = 0.3, CHCl_3_). IR (KBr): 2983, 2911, 1719 (C=O), 1574, 1553, 1439, 1396, 1285, 1240, 1136, 1063, 762. ^1^H-NMR (300 MHz, CDCl_3_): 0.90 (d, 1H, *J =* 9.5, H-7a), 0.98 (s, 3H, Me-8), 1.18 (s, 3H, Me-9), 1.50-1.63 (m, 1H, H-3a), 1.80–2.13 (m, 5H, H-5, H-4a, H-4b, H-1, H-3b), 2.30–2.46 (m, 2H, H-2, H-7b), 2.83–2.99 (m, 2H, H-10a, H-10b), 3.96 (s, 3H, Me-7′), 7.18 (dd, 1H, *J* = 7.7, 4.8, H-3′), 8.23 (d, 1H, *J* = 7.7, H-4′), 8.73 (d, 1H, *J* = 4.8, H-2′). ^13^C-NMR (75 MHz, CDCl_3_): 21.9 (C-3), 23.2 (C-8), 26.1 (C-4), 27.9 (C-9), 33.2 (C-7), 38.6 (C-6), 40.0 (C-2), 41.2 (C-5), 45.5 (C-1), 45.5 (C-10), 52.5 (C-7′), 119.8 (C-3′), 123.7 (C-5′), 139.0 (C-4′), 152.6 (C-2′), 161.8 (C-1′), 165.7 (C-6′). MS (ESI, 20 eV): 338.25 (100) [M+H]^+^. M = 337.50, C_17_H_23_NO_2_S_2_.

*((1S,2R,3S,5R)-3-(benzo[d]thiazol-2-yldisulfanyl)-6,6-dimethylbicyclo[3.1.1]heptan-2-yl)methanol (***3a***).* Yield: 78%. Yellowish oil. *R*_f_ = 0.24 (PhH/EtOAc, 10:1). ${\left[\alpha \right]}_{D}^{26}$ = +104.7 (*c* = 1.2, CHCl_3_). IR (KBr): 3383 (OH), 2924, 1460, 1425, 1238, 1049, 1005, 758, 729. ^1^H-NMR (300 MHz, CDCl_3_): 0.94 (s, 3H, Me-8), 1.17 (d, 1H, *J =* 10.3, H-7a), 1.22 (s, 3H, Me-9), 2.00 (tt, 1H, *J =* 5.6, 3.0, H-5), 2.11 (td, 1H, *J =* 5.9, 2.2, H-1), 2.20 (ddd, 1H, *J =* 13.9, 5.9, 2.9, H-4a), 2.33 (qd, 1H, *J =* 7.3, 2.2, H-2), 2.39–2.50 (m, 1H, H-7b), 2.57–2.73 (m, 2H, H-4b, OH), 3.46–3.54 (m, 1H, H-3), 3.68 (dd, 1H, *J* = 10.6, 7.7, H-10a), 3.89 (dd, 1H, *J* = 10.6, 7.3, H-10b), 7.34 (t, 1H, *J* = 7.8, H-5′), 7.45 (t, 1H, *J* = 7.8, H-4′), 7.80 (d, 1H, *J* = 7.8, H-6′), 7.91 (d, 1H, *J* = 7.8, H-3′). ^13^C-NMR (75 MHz, CDCl_3_): 23.4 (C-8), 27.3 (C-9), 32.5 (C-7), 36.7 (C-4), 38.5 (C-6), 41.8 (C-5), 43.1 (C-1), 44.3 (C-3), 52.1 (C-2), 66.4 (C-10), 121.2 (C-6′), 122.0 (C-3′), 124.7 (C-5′), 126.3 (C-4′), 135.8 (C-7′), 154.3 (C-2′), 171.9 (C-1′). MS (ESI, 20 eV): 352.16 (29) [M+H]^+^. M = 351.54, C_17_H_21_NOS_3_.

*((1S,2R,3S,5R)-3-((4,6-dimethylpyrimidin-2-yl)disulfanyl)-6,6-dimethylbicyclo[3.1.1]heptan-2-yl)methanol (***3b***).* Yield: 57%. Yellowish oil. *R*_f_ = 0.26 (PhH/EtOAc, 5:1). ${\left[\alpha \right]}_{D}^{26}$ = +54.9 (*c* = 0.7, CHCl_3_). IR (KBr): 3416 (OH), 2922 1583, 1529, 1437, 1254, 1047, 879, 756, 546. ^1^H-NMR (300 MHz, CDCl_3_): 0.94 (s, 3H, Me-8), 1.13 (d, 1H, *J =* 10.3, H-7a), 1.19 (s, 3H, Me-9), 1.84 (td, 1H, *J* = 5.9, 1.7, H-1), 1.97 (tt, 1H, *J =* 5.5, 2.9, H-5), 2.20 (ddd, 1H, *J* = 14.3, 6.2, 2.9, H-4a), 2.30 (dddd, 1H, *J* = 10.2, 7.6, 5.0, 2.2, H-2), 2.37–2.46 (m, 1H, H-7b), 2.49 (s, 6H, Me-5′, Me-6′), 2.61–2.72 (m, 1H, H-4b), 2.51–3.59 (m, 1H, H-3), 3.62 (dd, 1H, *J* = 11.0, 5.1, H-10a), 3.62 (t, 1H, *J* = 11.0, H-10b), 4.28 (br.s, 1H, OH), 6.83 (s, 1H, H-3′). ^13^C-NMR (75 MHz, CDCl_3_): 23.5 (C-8), 23.6 (C-5′, C-6′), 27.5 (C-9), 33.4 (C-7), 38.4 (C-4), 38.5 (C-6), 41.9 (C-5), 44.2 (C-1), 44.7 (C-3), 50.5 (C-2), 68.1 (C-10), 117.5 (C-3′), 168.0 (C-2′, C-4′), 170.9 (C-1′). MS (ESI, 20 eV): 325.21 (100) [M+H]^+^. M = 324.50, C_16_H_24_N_2_OS_2_.

*Methyl 2-(((1S,2R,3S,5R)-2-(hydroxymethyl)-6,6-dimethylbicyclo[3.1.1]hept-3-yl)disulfanyl)pyridine-3-carboxylate (**3c**).* Yield: 88%. Yellowish oil. R_f_ = 0.24 (PhH/EtOAc, 10:1). ${\left[\alpha \right]}_{D}^{26}$ = +192.1 (c = 1.1, CHCl_3_). IR (KBr): 3343 (OH), 2924, 1719 (C=O), 1572, 1557, 1443, 1396, 1286, 1244, 1138, 1062, 763, 729, 680. ^1^H-NMR (300 MHz, C_6_D_6_): 0.67 (s, 3H, Me-8), 0.97 (s, 3H, Me-9), 1.12 (d, 1H, *J* = 10.3, H-7a), 1.66–1.75 (m, 2H, H-5, H-1), 2.15–2.25 (m, 1H, H-7b), 2.35 (ddd, 1H, *J* = 14.1, 5.7, 2.2, H-4a), 2.44 (dddd, 1H, *J* = 9.6, 7.2, 5.1, 2.2, H-2), 2.52–2.63 (m, 1H, H-4b), 3.37 (s, 3H, Me-7′), 3.53–3.63 (m, 1H, H-3), 3.73 (dd, 1H, *J* = 10.3, 4.4, H-10a), 4.02 (t, 1H, *J* = 10.3, H-10b), 4.77 (br.s, 1H, OH), 6.32 (dd, 1H, *J* = 7.7, 4.8, H-3′), 7.68 (dd, 1H, *J* = 7.7, 1.7, H-4′), 8.45 (dd, 1H, *J* = 4.8, 1.7, H-2′). ^13^C-NMR (75 MHz, CDCl_3_): 23.9 (C-8), 27.7 (C-9), 33.9 (C-7), 38.7 (C-6), 39.0 (C-4), 42.7 (C-5), 45.0 (C-1), 45.1 (C-3), 52.2 (C-7′), 52.7 (C-2), 68.7 (C-10), 120.4 (C-3′), 124.9 (C-5′), 139.5 (C-4′), 152.8 (C-2′), 162.6 (C-1′), 165.5 (C-6′). MS (ESI, 20 eV): 354.21 (23) [M+H]^+^. M = 353.50, C_17_H_23_NO_3_S_2_.

*(1S,2S,3S,5R)-2-((benzo[d]thiazol-2-yldisulfanyl)methyl)-6,6-dimethylbicyclo[3.1.1]heptan-3-ol (***4a***).* Yield: 68%. Yellowish oil. *R*_f_ = 0.42 (PhH/EtOAc, 5:1). ${\left[\alpha \right]}_{D}^{27}$ = +29.9 (*c* = 0.4, CHCl_3_). IR (KBr): 3377 (OH), 1460, 1427, 1275, 1236, 1038, 1005, 758. ^1^H-NMR (300 MHz, C_6_D_6_): 0.59 (s, 3H, Me-8), 1.04 (s, 3H, Me-9), 1.17 (d, 1H, *J =* 9.5, H-7a), 1.78 (tt, 1H, *J =* 5.5, 2.9, H-5), 1.79–1.91 (m, 2H, H-1, H-4a), 2.15–2.23 (m, 1H, H-7b), 2.30–2.39 (m, 2H, H-4b, H-2), 3.50 (m, 1H, H-3), 2.75 (dd, 1H, *J* = 13.2, 6.6, H-10a), 2.97 (dd, 1H, *J* = 13.2, 8.8, H-10b), 3.17 (br.s, 1H, OH), 4.01 (dt, 1H, *J* = 9.2, 4.2, H-3), 6.91 (t, 1H, *J* = 7.8, H-5′), 7.07 (t, 1H, *J* = 7.8, H-4′), 7.28 (d, 1H, *J* = 7.8, H-6′), 7.96 (d, 1H, *J* = 7.8, H-3′). ^13^C-NMR (75 MHz, CDCl_3_): 24.0 (C-8), 27.6 (C-9), 33.4 (C-7), 38.3 (C-6), 39.0 (C-4), 42.0 (C-5), 45.9 (C-1), 46.2 (C-10), 53.1 (C-2), 69.1 (C-3), 121.6 (C-6′), 123.0 (C-3′), 125.2 (C-5′), 126.9 (C-4′), 136.8 (C-7′), 155.5 (C-2′), 171.7 (C-1′). MS (ESI, 20 eV): 352.43 (100) [M+H]^+^. M = 351.54, C_17_H_21_NOS_3_.

*(1S,2S,3S,5R)-2-(((4,6-dimethylpyrimidin-2-yl)disulfanyl)methyl)-6,6-dimethylbicyclo[3.1.1]heptan-3-ol (***4b***).* Yield: 48 %. Yellowish oil. *R*_f_ = 0.35 (PE/EtOAc, 2:1). ${\left[\alpha \right]}_{D}^{27}$ = +153.5 (*c* = 0.6, CHCl_3_). IR (KBr): 3343 (OH), 2983, 2918, 1585, 1530, 1433, 1366, 1344, 1294, 1256, 1032, 756. ^1^H-NMR (300 MHz, CDCl_3_): 0.86 (s, 3H, Me-8), 1.12 (d, 1H, *J =* 9.5, H-7a), 1.19 (s, 3H, Me-9), 1.77–1.86 (m, 2H, H-1, H-4a), 1.90–2.00 (m, 1H, H-5), 2.15–2.25 (m, 1H, H-2), 2.27–2.37 (m, 1H, H-7b), 2.47 (s, 6H, Me-5′, Me-6′), 2.46–2.59 (m, 1H, H-4b), 2.83–3.03 (m, 2H, H-10a, H-10b), 4.14 (br.s, 1H, OH), 4.28 (dt, *J* = 8.6, 4.2, 1H, H-3), 6.80 (s, 1H, H-3′). ^13^C-NMR (75 MHz, CDCl_3_): 23.6 (C-5′, C-6′), 23.8 (C-8), 27.0 (C-9), 32.5 (C-7), 37.3 (C-4), 37.8 (C-6), 41.2 (C-5), 46.6 (C-1), 46.7 (C-10), 52.6 (C-2), 68.6 (C-3), 117.4 (C-3′), 168.0 (C-2′,C-4′), 170.3 (C-1′). MS (ESI, 20 eV): 325.21 (100) [M+H]^+^. M = 324.50, C_16_H_24_N_2_OS_2_.

*Methyl 2-((((1S,2S,3S,5R)-3-hydroxy-6,6-dimethylbicyclo[3.1.1]heptan-2-yl)methyl)disulfanyl)pyridine-3-carboxylate (**4c**)*. Yield: 85%. Yellowish gum. R_f_ = 0.25 (PE/EtOAc 3:1). ${\left[\alpha \right]}_{D}^{27}$ = +118.7 (c = 1.0, CHCl_3_). IR (KBr): 3400 (OH), 1715 (C=O), 1572, 1557, 1398, 1288, 1244, 1138, 1065, 762. ^1^H-NMR (300 MHz, CDCl_3_): 0.87 (s, 3H, Me-8), 1.10 (d, 1H, *J* = 9.5, H-7a), 1.19 (s, 3H, Me-9), 1.77–1.86 (m, 2H, H-4a, H-1), 1.90–1.99 (m, 1H, H-5), 2.14–2.22 (m, 1H, H-2), 2.27–2.37 (m, 1H, H-7b), 2.46–2.59 (m, 1H, H-4b), 2.82–3.04 (m, 2H, H-10a, H-10b), 3.96 (s, 3H, Me-7′), 4.32 (dt, 1H, *J* = 9.0, 4.0, H-3), 4.86 (br.s, 1H, OH), 7.22 (dd, 1H, *J* = 7.7, 4.8, H-3′), 8.26 (d, 1H, *J* = 7.7, H-4′), 8.66 (d, 1H, *J* = 4.8, H-2′). ^13^C-NMR (75 MHz, CDCl_3_): 23.8 (C-8), 27.1 (C-9), 32.8 (C-7), 37.3 (C-4), 37.9 (C-6), 41.3 (C-5), 46.3 (C-10), 46.6 (C-1), 52.6 (C-2, C-7′), 69.1 (C-3), 120.3 (C-3′), 123.7 (C-5′), 139.4 (C-4′), 152.3 (C-2′), 161.4 (C-1′), 165.4 (C-6′). MS (ESI, 20 eV): 354.21 (100) [M+H]^+^. M = 353.50, C_17_H_23_NO_3_S_2_.

#### 4.2.2. General Method for the Synthesis of Carboxylic Acids

The method is based on the procedure [50].

A solution of 0.90 mmol of LiOH in 3–4 mL of H_2_O was added with stirring to a solution of 0.45 mmol of substrate **1c–4c** in 6 mL of THF and left under stirring for 4–5 h. The progress of the reaction was monitored by TLC. Upon completion of the reaction, the solvent was removed under reduced pressure, the residue was diluted with H_2_O (15–20 mL), neutralized with 2 M HCl to pH = 4, extracted with CHCl_3_. The organic layer was dried over Na_2_SO_4_, the solvent was removed under reduced pressure. The product was purified by recrystallization from hexane-Et_2_O system.

*2-(((1S,2S,5R)-2-isopropyl-5-methylcyclohexyl)disulfanyl)nicotinic acid (***1d***).* Yield: 80%. White powder. M.p. 160–161 °C with decomposition. ${\left[\alpha \right]}_{D}^{27}$ = +147.2 (*c* = 0.180, CHCl_3_). IR (KBr): 3080, 3028, 2647, 2912, 2870, 2841, 2650, 2549, 1684, 1571, 1552, 1446, 1419, 1386, 1292, 1251, 1149, 1126, 1060, 916, 821, 763, 713, 557. ^1^H-NMR (300 MHz, CDCl_3_): 0.85 (d, 3H, *J* = 6.16, Me-7), 0.90–1.00 (m, 1H, H-4a), 0.95 (d, 3H, *J* = 6.64 Hz, Me-10), 1.03–1.27 (m, 2H, H-2, H-6), 1.20 (d, 3H, *J* = 2.64 Hz, Me-9), 1.33 (td, 1H, *J* = 12.69, 3.08 Hz, H-3a), 1.64–1.99 (m, 3H, H-3b, 4b, H-5), 2.04–2.24 (m, 2H, H-6, H-8), 3.35 (d, 1H, *J* = 1.76 Hz, H-1), 7.18–7.26 (m, 1H, H-3′), 8.35 (dd, 1H, *J* = 7.78, 1.61 Hz, H-4′), 8.73 (dd, 1H, *J* = 4.70, 1.47 Hz, H-2′). ^13^C-NMR (75 MHz, CDCl_3_): 20.89 (C-10), 21.35 (C-9), 22.09 (C-7), 25.99 (C-3), 26.43 (C-5), 30.22 (C-8), 35.48 (C-4), 39.42 (C-6), 49.21 (C-2), 53.14 (C-1), 119.91 (C-3′), 122.84 (C-5′), 139.93 (C-4′), 152.99 (C-2′), 163.64 (C-1′), 170.30 (C-6′). MS (ESI, 20 eV): 326.07 (100) [M+H]^+^. M = 325.47, C_16_H_23_NO_2_S_2_.

*2-((((1S,2R,5S)-6,6-dimethylbicyclo[3.1.1]heptan-2-yl)methyl)disulfanyl)nicotinic acid (***2d***).* Yield: 85%. White powder. M.p. 150 with decomposition. ${\left[\alpha \right]}_{D}^{27}$ = −63.3 (*c* = 0.139, CHCl_3_). IR (KBr): 3041, 2980, 2941, 2910, 2868, 2650, 2553, 1691, 1571, 1552, 1465, 1429, 1388, 1286, 1255, 1230, 1145, 1064, 904, 815, 759, 715, 555. ^1^H-NMR (300 MHz, CDCl_3_): 0.84 (d, 1H, *J* = 9.68, H-7a), 0.92 (s, 3H, Me-8), 1.12 (s, 3H, Me-9), 1.42–1.60 (m, 1H, H-3a), 1.70–2.08 (m, 5H, H-1, H-3b, H-4a, H-4b, H-5), 2.20–2.42 (m, 2H, H-2, H-7b), 2.72–2.95 (m, 2H, H-10a, 10b), 7.14 (dd, 1H, *J* = 7.63, 4.70 Hz, H-3′), 8.23 (dd, 1H, *J* = 7.63, 1.47 Hz = H-4′), 8.66 (dd, 1H, *J* = 4.55, 1.61 Hz, H-2′). ^13^C-NMR (75 MHz, CDCl_3_): 21.73 (C-3), 23.03 (C-8), 25.99 (C-4), 27.75 (C-9), 33.07 (C-7), 38.53 (C-6), 39.94 (C-2), 41.12 (C-5), 44.37 (C-1), 45.40 (C-10), 119.81 (C-3′), 124.08 (C-5′), 139.45 (C-4′), 152.36 (C-2′), 161.73 (C-1′), 167.27 (C-6′). MS (ESI, 20 eV): 324.40 (100) [M+H]^+^. M = 323.47, C_16_H_21_NO_2_S_2_.

*2-(((1S,2R,3S,5R)-2-(hydroxymethyl)-6,6-dimethylbicyclo[3.1.1]heptan-3-yl)disulfanyl)nicotinic acid (***3d***).* Yield: 95%. Yellow powder. M.p. 99 °C. ${\left[\alpha \right]}_{D}^{27}$ = −225.3 (*c* = 0.141, CHCl_3_). IR (KBr): 3238 (OH), 2985, 2926, 1707 (C=O), 1560, 1390, 1271, 1246, 1141, 1062, 1041, 808, 765, 704, 547. ^1^H-NMR (300 MHz, CDCl_3_): 0.91 (s, 2H, Me-8), 1.14 (d, 1H, *J* = 9.98 Hz, H-7a), 1.88 (t, 1H, *J* = 5.14 Hz, H-1), 1.94–2.05 (m, 1H, H-5), 2.22–2.51 (m, 3H, H-2, H-6a, H-7b), 2.70 (t, 1H, *J* = 12.03 Hz, H-6b), 3.47 (dt, 1H, *J* = 9.61, 6.79 Hz, H-3), 3.73 (dd, 1H, *J* = 10.86, 4.11 Hz, H-10a), 3.93 (t, 1H, *J* = 10.56 Hz, H-10b), 7.30 (dd, 1H, *J* = 7.78, 4.84 Hz, H-3′), 9.41 (dd, 1H, *J* = 7.92, 1.47 Hz, H-4′), 8.77 (dd, 1H, *J* = 4.70, 1.47 Hz, H-2′). ^13^C-NMR (75 MHz, CDCl_3_): 23.66 (C-8), 27.38 (C-9), 33.49 (C-7), 38.31 (C-4), 38.52 (C-6), 41.93 (C-5), 44.35 (C-1), 44.74 (C-3), 51.04 (C-2), 68.17 (C-10), 120.51 (C-3′), 124.17 (C-5′), 140.46 (C-4′), 152.75 (C-2′), 162.30 (C-1′), 168.31 (C-6′). MS (ESI, 20 eV): 340.12 (100) [M+H]^+^. M = 339.47, C_16_H_21_NO_3_S_2_.

*2-((((1S,2S,3S,5R)-3-hydroxy-6,6-dimethylbicyclo[3.1.1]heptan-2-yl)methyl)disulfanyl)nicotinic acid (***4d***).* Yield: 73%. White powder. m.p. 172–173 °C. ${\left[\alpha \right]}_{D}^{27}$ = +169.2 (*c* = 0.130, CHCl_3_). IR (KBr): 3256 (OH), 2985, 2920, 1703 (C=O), 1562, 1388, 1296, 1259, 1144, 1063, 1032, 810, 764, 704, 542. ^1^H-NMR (300 MHz, (CD_3_)O): 0.91 (s, 3H, Me-8), 1.09 (d, 1H, H-7a, *J* = 9.4 Hz), 1.77 (dt, 1H, H-4, *J* = 14, 3.3 Hz), 1.87–1.97 (m, 1H, H-5), 2.11–2.24 (m, 2H, H-1, 2), 2.31–2.42 (m, 1H, H-7), 2.42–2.55 (m, 1H, H-4), 2.88 (dd, 1H, H-10a, *J* = 12.9, 7.9 Hz), 3.14 (dd, 1H, H-140b, *J* = 12.9, 7.8 Hz), 4.15 (dt, 1H, H-3, *J* = 9, 4.4 Hz), 7.39 (dd, 1H, H-3′, *J* = 7.6, 4.7 Hz), 8.35 (d, 1H, H-4′, *J* = 7.9 Hz), 8.71 (d, 1H, H-2′, *J* = 4.7 Hz). ^13^C-NMR (75 MHz, (CD_3_)O): 24.03 (C-8), 27.84 (C-9), 33.67 (C-7), 38.87 (C-6), 33.19 (C-4), 42.58 (C-5), 45.48 (C-10), 45.98 (C-1), 53.52 (C-2), 69.31 (C-3), 121.51 (C-3′), 133.13 (C-5′), 140.39 (C-4′), 153.39 (C-2′), 168.78 (C-6′). MS (ESI, 20 eV): 340.14 (100) [M+H]^+^. M = 339.47, C_16_H_21_NO_3_S_2_.

### 4.3. Antioxidant Activity (Non-Cellular Model)

The antioxidant activity of monoterpenylhetaryl disulfides was evaluated for their in vitro ability to inhibit lipid peroxidation (LPO) processes in the brain lipids of laboratory mice [51,52]. The brain was removed and homogenized (10%) in physiological saline (pH = 7.4) and centrifuged for 10 min. Then, an aliquot of the supernatant (S1) was taken [51,53], into which the test compounds were added as solutions in ethanol (final concentration of 100 μM and 1 mM). After 30 min, LPO was initiated by adding a freshly prepared solution of FeSO_4_ and ascorbic acid [54]. Then, the test samples were incubated for one hour at 37 °C with slow stirring in a thermostatic Biosan ES-20 shaker (Biosan, Riga, Latvia). The concentration of LPO secondary products reacting with 2-thiobarbituric acid (thiobarbituric acid reactive substances, TBA-RS) was determined using ThermoSpectromic Genesys 20 spectrophotometer (Thermo Fisher Scientific, Waltham, MA, USA) at λ = 532 nm, the extinction coefficient 1.56 × 10^5^ M^−1^cm^−1^ was used for calculations [52,55].

### 4.4. Erythrotoxicity, Antioxidant Activity, Membrane-Protective Activity (Cellular Model System)

To evaluate the erythrotoxicity, antioxidant and membrane-protective activity of the compounds, we used 0.5% (*v*/*v*) suspension of red blood cells from laboratory mice in phosphate-buffered saline (PBS, pH 7.4). The test substances were previously dissolved in ethanol. Erythrotoxicity (hemolytic activity) was evaluated by the degree of hemolysis of red blood cells 1–5 h after the introduction of solutions of the compounds. Membrane-protective and antioxidant activity was determined by the degree of inhibition of H_2_O_2_-induced hemolysis. For this purpose, 30 min after the erythrocyte suspension was added to the solutions of the test compounds, hemolysis was initiated with a solution of hydrogen peroxide (0.006%). Then, the reaction mixture was incubated in a thermostatic shaker Biosan ES-20 with slow stirring at 37 °C for 5 h. An aliquot was taken from the incubation medium every hour, centrifuged for 5 min (1600× *g*), and the degree of hemolysis was determined by the hemoglobin content in the supernatant on a ThermoSpectromic Genesys 20 spectrophotometer (Thermo Fisher Scientific, Waltham, MA, USA) at 524 nm [56]. The percentage of hemolysis was calculated in relation to the complete hemolysis of the sample. Each experiment was designed with 4–8 replicates. Statistical data processing and charting were performed using the Microsoft Office Excel 2007 software package (Microsoft, Redmond, WA, USA). The data represent the mean ± SE.

### 4.5. Antibacterial Activity

Minimum inhibitory concentrations (MICs) of compounds was determined by the broth microdilution assay in 96-well plates (Eppendorf, Hamburg, Germany) according to the EUCAST rules for antimicrobial susceptibility testing [57] in full Mueller-Hinton broth (MH). The concentrations of compounds to be tested ranged from 1 to 2048 µg/mL. The MIC was determined as the lowest concentration of an antibiotic for which no visible bacterial growth could be observed after 24 h of incubation.

### 4.6. Antifungal Activity

MICs on *C. albicans* was determined using the broth microdilution method in 96-well plates (Eppendorf) in MH broth as recommended in the protocol CLSI M27-A3 [58]. *C. albicans* was grown overnight and diluted with MH broth until optical density of 0.5 to obtain the working solution. Then, 2-fold serial dilutions of compounds in concentrations from 1 to 1024 μg/mL were prepared in MH broth and seeded with fungi (1% *v*/*v* of working solution) with subsequent incubation at 37 °C for 24 h. The MIC was defined as the lowest concentration of the compound at which no visible growth could be seen.

### 4.7. Mutagenicity and Cytotoxicity

The mutagenicity of compounds was evaluated in the Ames test with *S. typhimurium* TA98, TA100 and TA102 strains as described in [38]. The spot-test modification has been applied to avoid false-negative results driven with antibacterial activity of compounds. The tested compound was considered to be mutagenic if the number of revertant colonies increased more than 2 times close to the filter paper with compound.

Cytotoxicity of compounds was determined using the microtetrazolium test (MTT) on EBL cells. The cells were cultured in DMEM—Dulbecco’s Modified Eagle’s Medium (Sigma Aldrich, St. Louis, MO, USA) supplemented with 10% FBS, 2 mM L-glutamine, 100 µg/mL penicillin and 100 µg/mL streptomycin. Cells were seeded in 96-well plates with the density of 3000 cells per well and left overnight to allow for the attachment. Cells were next cultured at 37 °C and 5% CO_2_ in the presence of compounds of interest at various concentrations from 1.25 to 160 µg/mL. After 24 h of cultivation the cells were subjected to MTT-assay. The formazan was solubilized by DMSO and the optical density was measured on Tecan Infinite 200Pro at 570 nm. The concentration required to inhibit cellular dehydrogenase activity by 50% (CC_50_ value) was calculated by using GraphPad Prism 6.0 software (GraphPad software, San Diego, CA, USA).

### 4.8. Antiviral Activity Assessment

#### 4.8.1. Toxicity Studies

MTT [59] was used to study cytotoxicity of the compounds. Briefly, series of three-fold dilutions of each compound (300–4 µg/mL) in MEM were prepared. MDCK cells (ATCC CCL-34) were incubated for 72 h at 36 °C in 5% CO_2_ in the presence of the dissolved substances. The degree of destruction of the cell monolayer was then evaluated in the MTT. The cells were washed twice with saline, and a solution of 3-(4,5-dimethylthiazol-2-yl)-2,5-diphenyltetrazolium bromide (0.5 mg/mL) in cell culture medium was added to the wells. After 1h incubation, the wells were washed and the formazan residue dissolved in DMSO (0.1 mL per well). The optical density of wells was then measured on a Multiscan FC photometer at wavelength of 540 nm and plotted against concentration of the compounds. Each concentration was tested in three parallels. The 50% cytotoxic dose (CC_50_) of each compound (i.e., the compound concentration that causes the death of 50% cells in a culture, or decreasing the optical density twice as compared to the control wells) was calculated from the data obtained. Values of CC_50_ obtained in µg/mL were then calculated into micromoles.

#### 4.8.2. Cell Protection Assay

The compounds in appropriate concentrations were added to MDCK cells (0.1 mL per well). Cells were further infected with A/Puerto Rico/8/34 (H1N1) influenza virus (m.o.i 0.01). Plates were incubated for 72 h at 36 °C at 5% CO_2_. After that, cell viability was assessed by MTT test as described above. The cytoprotective activity of compounds was considered as their ability to increase the values of IC_50_ compared to control wells (with virus only, no drugs). Based on the results obtained, the values of IC_50_, i.e., concentration of compounds that results in 50% cells protection, were calculated using GraphPad Prism software. Values of IC_50_ obtained in μg/mL were then calculated into μM. For each compound, the value of selectivity index (SI) was calculated as ratio of CC_50_ to IC_50_. The compounds with SI of 10 and higher were considered as active [39].

## Data Availability

Not applicable.

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
