# Peer review of "Synthesis and Biological Activity of Unsymmetrical Monoterpenylhetaryl Disulfides"

_molecules, 2022, doi:10.3390/molecules27165101_

Round 1

Reviewer 1 Report

1.    I recommend adding a description of the obtained results on the biological activity of the compounds.

2.    Figure 3 does not look very good, perhaps it is better to split it into 2 or remake it into a table.

3.    Table 1 is split into 2 pages, it would be better to move the text and place the whole table on another page.

4.    In table 2 change CC50 (subscript 50) and here compound 4 CC50 =38? then there is no standard deviation.

Author Response

Dear Reviewer!

Thank you for your comments and suggestions!

  1. I recommend adding a description of the obtained results on the biological activity of the compounds.

Response. The description of the obtained results on biological activity is given in the text of the manuscript, references to original sources and procedures are inserted in the experimental part.

  1. Figure 3 does not look very good, perhaps it is better to split it into 2 or remake it into a table.

Response. Figure 3 has been split into two parts.

  1. Table 1 is split into 2 pages, it would be better to move the text and place the whole table on another page.

Response. Table 1 has been moved.

  1. In table 2 change CC50 (subscript 50) and here compound 4 CC50 =38? then there is no standard deviation.

Response. The typos have been fixed.

Best regards,

Denis V. Sudarikov.

Reviewer 2 Report

                                 Review of the Article 

        Synthesis and biological activity of unsymmetrical mono-terpenylhetaryl disulfides. 

                                   By Denis V. Sudarikov at al,

   Paper describes the reaction of synthesies a series of unsymmetrical disulfides combining mono-  terpene and heterocyclic fragments.Disulfides with fragments of methyl esters 2-mercaptonicotinic acid was hydrolyzed to desired products. Yields for above two conversions are 48-88% and 73-95% respectivetly. This is interesting article,mainly considering new unsymmetrical monoterpenylhetaryl disulfides  synthesized for the first time.

   The synthetized products [compounds(1-4c) and (1-4d)]were evaluated for antioxidant, antibacterial, antifungal activity, cytotoxicity an mutagenicity.The results are promissing. This is the main part of reviewed Article.

   Remarks: In my opinion the following parts should be changed.

   Line 109. Authors should add "according to the procedure described earlier by another researchers"(you are obligated    to find it).

   Line 117. If Authors confirmed the structure and configuration of 1c,the question is:Which diastereoisomer you obtained? For three stereogenic carbon atoms in terpenethiol moiety there are possible eight diferent diastereoisomers.Add the sentence:  "Methyl 2-[[(1S,2S,5R)-2-isopropyl-5methylcyclohexyl]disulfanyl]pyridine-3-carboxylate(1c)"

   Line 319. Also in General procedure for the synthesis of unsymmetrical disulfides should be added reference of oryginal paper describing synthesis of unsymetrical disulfides(the same reference  like in line 109).

   Line 322. Insted of "stirred in the cold for 5 min.",should be "stirred at temperature 0*C for 5 min."

   Line 325. Insted of "the ice was removed",should be "the ice bath was removed".

   The chemistry presented here is interesting,although not novel.

Author Response

Dear Reviewer!

Thank you for your comments and suggestions!

Paper describes the reaction of synthesies a series of unsymmetrical disulfides combining mono-  terpene and heterocyclic fragments.Disulfides with fragments of methyl esters 2-mercaptonicotinic acid was hydrolyzed to desired products. Yields for above two conversions are 48-88% and 73-95% respectivetly. This is interesting article,mainly considering new unsymmetrical monoterpenylhetaryl disulfides  synthesized for the first time.

The synthetized products [compounds(1-4c) and (1-4d)]were evaluated for antioxidant, antibacterial, antifungal activity, cytotoxicity an mutagenicity.The results are promissing. This is the main part of reviewed Article.

Remarks: In my opinion the following parts should be changed.

Line 109. Authors should add "according to the procedure described earlier by another researchers"(you are obligated to find it).

Response. References to the original procedures have been added.

Line 117. If Authors confirmed the structure and configuration of 1c,the question is:Which diastereoisomer you obtained? For three stereogenic carbon atoms in terpenethiol moiety there are possible eight diferent diastereoisomers.Add the sentence:  "Methyl 2-[[(1S,2S,5R)-2-isopropyl-5methylcyclohexyl]disulfanyl]pyridine-3-carboxylate(1c)"

Response. Thank you. We have added the full IUPAC name to show the configuration of the chiral centers.

Line 319. Also in General procedure for the synthesis of unsymmetrical disulfides should be added reference of oryginal paper describing synthesis of unsymetrical disulfides(the same reference  like in line 109).

Response. References to the original procedures have been added.

Line 322. Insted of "stirred in the cold for 5 min.", should be "stirred at temperature 0*C for 5 min."

Response. Thank you. Your comments have been accepted and corrected.

Line 325. Insted of "the ice was removed", should be "the ice bath was removed".

Response. Your comments have been accepted and corrected.

The chemistry presented here is interesting,although not novel.

Thank you!

Best regards,

Denis V. Sudarikov.
